# Emergency care knowledge, utilization, and barriers in Northern Tanzania: A community-based survey

Frida Shayo[1,2,3], Gregory Goodluck Zaccheus[2,4], Francis Sakita[1,2,5], Thiago Rocha Hernandes[6], Joao Ricardo Nickenig Vissoci[5,7,8], Alexander Gordee[9], Maragatha Kuchibhatla[9], Michael Kiremeji[10], Linda Minja[1,3], Blandina T. Mmbaga[1,2,3,5], Catherine A. Staton[5,7,8], Elizabeth M. Keating[11], Anjni P. Joiner[5,7,8]*

1 Kilimanjaro Christian Medical Center, Moshi, Tanzania, 2 Kilimanjaro Christian Medical University College, Moshi, Tanzania, 3 Kilimanjaro Clinical Research Institute, Moshi, Tanzania, 4 Nuffield Department of Population Health, University of Oxford, Oxford, United Kingdom, 5 Global Emergency Medicine Innovation and Implementation (GEMINI) Research Center, Duke University, Durham, North Carolina, United States of America, 6 Department of Health Analysis, Metrics and Evidence, Pan American Health Organization, Washington, DC, United States of America, 7 Duke Global Health Institute, Duke University, Durham, North Carolina, United States of America, 8 Department of Emergency Medicine, Duke University, Durham, North Carolina, United States of America, 9 Department of Biostatistics, Duke University School of Medicine, Durham, North Carolina, United States of America, 10 Emergency Preparedness and Response, Ministry of Health, Dodoma, Tanzania, 11 Division of Pediatric Emergency Medicine, University of Utah, Salt Lake City, Utah, United States of America

* anjni.joiner@duke.edu

## Abstract

### Background

Emergency care systems are critical to improving care for time-sensitive emergency conditions. The growth and development of these systems in Sub-Saharan Africa is becoming a priority. Layperson knowledge and recognition of emergency symptoms and subsequent care-seeking behavior are key to achieving timely access to care and appropriate treatment. This study aimed to assess community knowledge of emergency conditions as well as barriers to accessing the emergency care system in Northern Tanzania.

### Methods

This was a cross-sectional study of households in three districts in Kilimanjaro, Tanzania from June to September 2021. The primary outcome was an inappropriate response to any of five hypothetical emergency conditions. Secondary outcomes were the incidence of household emergencies and delay in care access for those with emergency conditions. Data were analyzed using descriptive statistics. Associations between the outcome of interest and select household characteristics were analyzed using Fisher's Exact tests for categorical measures and Wilcoxon rank-sum tests for continuous measures.

### Results

A total of 539 households were interviewed with 2,274 participants. The majority (46.8%) were from Moshi District Council. 73.7% used cash and/or had no insurance. The mean

**Data availability statement:** Data are only available upon reasonable request. Data transfer requires a written agreement approved by Kilimanjaro Christian Medical Centre Ethics Committee and the National Institute for Medical Research (Tanzania). Data inquiries can be sent to Beatrice Temba (beatrice. temba@kcmuco.ac.tz).

**Funding:** This research was funded by the Josiah Charles Trent Memorial Foundation (APJ) and Duke Global Health Institute, Duke University (APJ). EMK receives funding from the Eunice Kennedy Shriver National Institute of Child Health and Human Development (grant number K23 HD112548). The funders had no role in study design, data collection and analysis, decision to publish, or preparation of the manuscript.

**Competing interests:** The authors have declared that no competing interests exist.

monthly household income was 226,107.6 Tanzanian Shillings. 76 (14.1%) households reported experiencing an emergency condition in the past year and 225 (41.7%) of respondents had an inappropriate response to at least one hypothetical emergency condition. A higher proportion of those with delayed access to healthcare paid with personal cash and a lower proportion had national health insurance. A higher proportion of those with inappropriate responses to hypothetical emergency conditions lived in rural districts, were uninsured, and had a lower mean income.

## Conclusion

Community-dwelling adults in Northern Tanzania have significant gaps in understanding of emergency care conditions and delayed access to care for these conditions. Distance to the healthcare facilities, cost, and lack of insurance may contribute to care delays. Increasing insurance coverage and developing emergency medical services may improve access to care.

## Introduction

Emergency conditions contribute substantially to the worldwide deaths, with Africa sharing a disproportionate burden [1]. Although the focus of emergency conditions has been primarily on communicable diseases and trauma, more than 80% of global deaths are due to non-communicable diseases (NCDs), with over half occurring in low- and middle-income countries (LMICs) [2]. According to the World Health Organization, non-communicable diseases increased from 4 to 7 of the global top ten causes of death in 2019, with ischemic heart disease and stroke as the leading contributors to mortality [2]. Tanzania, like other LMICs, is facing a tremendous burden of NCDs which account for approximately 34% of all deaths. Cardiovascular diseases alone account for 12% and injuries 11% [3]. Recognition of emergency symptoms at the community level with proper emergency care system activation and response has the potential to reduce the morbidity and mortality of NCDs significantly.

Emergency care systems encompass vital elements of the health system that link those experiencing acute illness and injuries with immediate care. These emergency care systems are composed of materials, infrastructure, the human workforce, and policies and procedures that allow care to be delivered in an organized, efficient manner [4]. While emergency care is gradually gaining ground in Sub-Saharan Africa, further development of emergency care systems throughout the continent is still underway [5]. Recognizing the importance of timely care for the acutely ill and injured, the World Health Assembly placed a top priority on strengthening emergency care systems for universal health coverage in May 2019, urging member state governments to take action [6]. By establishing high-quality emergency care systems, we can potentially avert half of all deaths in LMICs [3,7].

Critical to the emergency care system is the level of knowledge among laypersons in recognizing emergency conditions. Evidence shows that communities with a high level of awareness in recognizing emergency symptoms have better outcomes [8]. Some areas of Tanzania are still facing challenges in accessing prompt care for emergency conditions, with several factors contributing to delays. In a study done using the three-delays model to describe pediatric injury care seeking in Northern Tanzania, delays in the decision to seek care included emergency recognition, applying first aid and traditional medicine at home, and anticipated challenges to seeking care including place and time of injury [9]. The three-delays model, originally developed for obstetric emergencies, has been applied to emergency conditions

(Delay 1: decision to seek care; Delay 2: reaching an appropriate facility; Delay 3: receipt of adequate care at the facility) [10].

There is limited information regarding knowledge levels and the obstacles people face when seeking appropriate emergency care at the community level in the Kilimanjaro region of Tanzania. This study aimed to assess the existing comprehension and utilization of emergency care services in the Kilimanjaro region. By understanding the current state of knowledge and care-seeking behavior for emergency conditions in this region, we can identify potential strategies to overcome barriers with interventions that enhance the provision of emergency care in Tanzania.

## Methods

### Study design and setting

This community-based, cross-sectional study uses household surveys performed in three districts in the Kilimanjaro region of Tanzania: Moshi District Council, Moshi Municipal Council, and Hai District Council. Moshi Municipal Council has an area of 63.4 km$^2$ and a population density of 2,907 persons/km$^2$. It is surrounded by Moshi District Council and Hai District Council, which have areas of 1,300 km$^2$ and 902 km$^2$ and population densities of 358.9 persons/km$^2$ and 233.3 persons/km$^2$, respectively [11]. The three districts share a total of 13 hospitals, with seven in Moshi Municipal Council, five in Moshi District Council, and two in Hai District Council. Additionally, Moshi Municipal Council has a single private tertiary care facility acting as a zonal referral hospital, Kilimanjaro Christian Medical Centre (KCMC), and a government regional hospital, Mawenzi Hospital. The 13 hospitals include health centers, district, regional and zonal hospitals. At the time when the study was conducted, well-established emergency care services were only provided at the zonal referral hospital. The rest provided inpatient and outpatient services together with maternal and pediatric services.

The average life expectancy in Tanzania is 67.3 years and per capita health expenditure is USD 35. Emergency conditions such as neonatal conditions, lower respiratory infections, diarrheal diseases, and road traffic incidents are leading causes of death [2]. In 2021, the annual government minimum wage was 300,000 Tanzanian Shillings (TZS). For non-government jobs, the minimum wage was variable. For example, for the agricultural services sector was 100,000 Tanzanian Shillings (TZS) per month, whereas the minimum for the health services sector was 135,000 TZS per month [12].

### Study population

Any individual living in Moshi District Council, Moshi Municipal Council, or Hai District Council was eligible for participation. Community-dwelling adults aged 18 or greater who spoke Kiswahili fluently were included. Those who declined participation were excluded.

### Sample size

The sample size was calculated using a 35% expected proportion of participants having experienced at least one emergency condition in the last year, based on a similar study in Cameroon [13]. With a precision of 5%, a confidence interval of 95%, and a design effect of 1.5, we estimated a minimum of 49 villages with a cluster size of 11 households per village and a total of 539 households surveyed.

### Sampling techniques

We used a cluster sampling methodology to obtain a representative population sample. In the initial stage, 49 villages were identified using population-weighted sampling. Once villages

were identified, we sampled 11 randomly selected households within the village. A random starting point was generated in each village using QGIS 2.18.7. A 1 km$^2$ polygon was placed around this starting point and 20 random points were generated within this polygon.

The study team consisted of a research coordinator and three research assistants. Prior to any data collection or recruitment, the research coordinator visited the ward and village leaders for each village to explain the aims of the study, obtain permission to perform household surveys, and provide evidence of ethical clearance. The research coordinator used QGIS 2.18.7 to identify a random point within each village from which we created a 1 km$^2$ polygon. Twenty random points were generated within the polygon and the closest dwelling meeting the definition of household to each point was approached by the research coordinator, who was accompanied by the village leader. Nine households were back-ups in case any of the first 11 were unavailable. This approach was used for 30 villages but was deemed to be challenging due to lack of current census data and subsequent changes within villages over time and thus required team members to walk lengthy distances to reach households. For villages 31 through 49, we employed an alternate technique to randomly identify households. Using publicly available satellite imagery, we used a building footprint technique to identify man-made structures within each village. Using ArcGIS Pro software, the GPS (geographical positioning system) coordinates of 20 randomly selected structures were identified.

In each approach, if the dwelling met the definition of a household (a single individual or group of persons living together who make common provisions for food and other essentials for living), the research coordinator approached any adult member of the household in order to explain the aims of the study. For any structure consisting of multiple stories or units, the bottom-floor unit or first household was selected. If the member(s) of the household expressed interest, the coordinates of the household were recorded on a Garmin eTrex hand-held GPS device and they were informed of an intent to return with the study team to conduct the survey.

Research assistants returned on a later date to the villages to obtain informed consent and conduct the interviews. Informed consent was obtained from the head of household or health care decision-maker in the household. Participating households were provided compensation in the form of a phone voucher worth 5,000 Tanzanian Shillings (2.16 USD). Verbal and written informed consent was obtained for all participants. Research assistants read the informed consent to the participants in the native Kiswahili language to ensure their comprehension given variable literacy rates.

## Data collection and measurements

Data collection was from June 1, 2021, to September 1, 2021. The head of household or health care decision-maker responded to survey questions for all household members. Study data were collected and managed using REDCap electronic data capture tools hosted at Kilimanjaro Clinical Research Institute [14]. When internet connectivity was unavailable, backup paper copies of the survey were used. Data quality checks were performed weekly by investigators and the study team and any missing information was reviewed with the research assistants.

A structured survey was developed with local Tanzanian emergency physician input (S1 File). The survey included sections on demographics and socioeconomics and emergency conditions. Given the multi-purpose use of the survey, additional questions were asked regarding alcohol use, food insecurity, and snake bites. Participants were asked for detailed information on age, gender, and medical history of household members, including which family members had experienced health emergencies in the past year and delays and barriers to access to care.

Five theoretical emergency scenarios were posed to the family members being surveyed (Table 1). All five scenarios were considered severe medical or traumatic (physical) situations requiring immediate care. These scenarios were chosen to reflect a wide variety of emergency

**Table 1. Emergency scenarios used in the survey.**

| | |
|---|---|
| Scenario 1 | Your older mother wakes up and can not move her right arm or leg. Her face is drooping and her speech is slurred. |
| Scenario 2 | Your 14-year-old son has very bad pain in his stomach and can not stop vomiting and crying. He has not eaten in 3 days and refuses water for 2 days. |
| Scenario 3 | Your 4-month-old has diarrhea, vomiting and very high fever for 2 days. She is sleeping all day, not drinking milk, and now you can not wake her up. |
| Scenario 4 | Your father wakes up with very bad chest pain and trouble breathing. |
| Scenario 5 | You are in a bad motorcycle crash and hit your head. Your left arm is broken and you can not breathe well. |

conditions, reflecting CD, NCDs, and traumatic conditions that are commonly seen in this region. They were reviewed and approved by Tanzanian and US-based emergency physicians for accuracy and research team members for readability. An inappropriate response to an emergency scenario was defined as someone who satisfied any of the 3 following situations for any of the 5 presented emergency scenarios (S1):

1. Someone stated they would not seek care

2. Classified as anything other than "severe" or "very severe"

3. Stated they would seek care "less than a day", "between 1–2 days" or "3 days or more"

## Main outcomes

The primary outcome of interest was a delay in reaching healthcare services among those households who experienced one or more health emergencies. A delay in reaching healthcare services was defined as either a lack of seeking care or a delay in arrival at the final location of care of more than one hour from the time of initially seeking care (Delays 1 and 2 in the three delays model). One hour was chosen as the cut-off point through a combination of literature around several emergency conditions [15,16].

Secondary outcomes included the prevalence of emergency conditions in the Kilimanjaro region, and the ability to recognize and appropriately respond to pre-specified hypothetical healthcare emergencies.

## Statistical analysis

Household demographics and characteristics were summarized using either mean with standard deviation or median with 25th and 75th percentiles for continuous measures and frequency with percentage for categorical measures, stratified by the occurrence of one or more healthcare emergencies. Responses to each of the five hypothetical emergency situations were similarly summarized. The association of select household characteristics and an inappropriate response to any of the five scenarios were assessed using Chi-square tests for categorical measures and Wilcoxon rank-sum tests for continuous measures.

Among those households with healthcare emergencies, emergency and care-seeking behaviors and characteristics were summarized using descriptive statistics, stratified by delayed or no care. Within this subset of households, the association between a delay in care access and select household characteristics of interest was examined using Fisher's Exact tests for categorical measures (similar to Chi-square tests but valid for smaller sample sizes) and Wilcoxon rank-sum tests for continuous measures.

All statistical analyses were performed using SAS 9.4 (SAS Institute Inc., Cary, NC). All statistical tests were two-sided and performed without adjustment for multiple testing.

### Ethical statements

This study received approval from the Duke Institutional Review Board (Pro000106116) research ethical clearance from Kilimanjaro Christian Medical University College (Research Proposal No. 1264) and ethical clearance from the National Institute for Medical Research, Tanzania.

## Results

A total of 539 households were interviewed with a total of 2,274 participants (Table 2). The majority (46.8%) were from Moshi District Council with a median household size of 4.0 (IQR 3.0–6.0) and median household income of 150,000 (90,000–300,000) Tanzanian shillings per month, approximately 64 USD (2023). The majority (N = 474, 88.1%) lived on non-paved

**Table 2. Household demographics by emergency care incidence in the previous 12 months.**

| | No emergency care incidence (N = 463) | Emergency care incidence (N = 76) | Total (N = 539) |
|---|---|---|---|
| **District of residence** | | | |
| Hai | 114 (24.6%) | 29 (38.2%) | 143 (26.5%) |
| Moshi municipal | 126 (27.2%) | 18 (23.7%) | 144 (26.7%) |
| Moshi district | 223 (48.2%) | 29 (38.2%) | 252 (46.8%) |
| **Tribe** | | | |
| Chagga | 323 (69.8%) | 56 (73.7%) | 379 (70.3%) |
| Sukuma | 7 (1.5%) | 0 (0.0%) | 7 (1.3%) |
| Muha | 4 (0.9%) | 0 (0.0%) | 4 (0.7%) |
| Sambaa | 12 (2.6%) | 2 (2.6%) | 14 (2.6%) |
| Iraq | 3 (0.6%) | 1 (1.3%) | 4 (0.7%) |
| Masaai | 6 (1.3%) | 0 (0.0%) | 6 (1.1%) |
| Nyaturu | 3 (0.6%) | 0 (0.0%) | 3 (0.6%) |
| Pare | 44 (9.5%) | 7 (9.2%) | 51 (9.5%) |
| Mmeru | 2 (0.4%) | 0 (0.0%) | 2 (0.4%) |
| Other | 59 (12.7%) | 10 (13.2%) | 69 (12.8%) |
| **Religious affiliation** | | | |
| Christian | 378 (81.6%) | 62 (81.6%) | 440 (81.6%) |
| Muslim | 78 (16.8%) | 12 (15.8%) | 90 (16.7%) |
| Hindu | 2 (0.4%) | 1 (1.3%) | 3 (0.6%) |
| Other/None | 5 (1.1%) | 1 (1.3%) | 6 (1.1%) |
| **Number of household members** | | | |
| Median (Q1, Q3) | 4 (3, 6) | 5 (3, 6) | 4 (3, 6) |
| **Monthly household income** | | | |
| Median (Q1, Q3) | 150000 (85000, 300000) | 200000 (100000, 310000) | 150000 (90000, 300000) |
| **Residence on paved road*** | 53 (11.5%) | 11 (14.5%) | 64 (11.9%) |
| **Residence reachable by 4-wheel vehicle** | | | |
| Can be reached by car | 376 (81.2%) | 70 (92.1%) | 446 (82.7%) |
| Can only be reached by foot | 87 (18.8%) | 6 (7.9%) | 93 (17.3%) |
| **Insurance status household member** | | | |
| None | 30 (6.5%) | 8 (10.5%) | 38 (7.1%) |
| Cash personal | 316 (68.3%) | 43 (56.6%) | 359 (66.6%) |
| National Health Insurance | 111 (24.0%) | 22 (28.9%) | 133 (24.7%) |
| Other | 6 (1.3%) | 3 (3.9%) | 9 (1.7%) |

*Missing 1 value.

roads but 446 (82.7%) households could be reached by four-wheel vehicle. Only 133 (24.7%) households were insured by National Health Insurance. Seventy-six (14.1%) households reported that a member of their household had experienced an emergency condition in the past year and 225 (41.7%) respondents had an inappropriate response to at least one hypothetical emergency condition (would not seek care, not classified as "severe" or "very severe", or would seek care "less than a day", "between 1–2 days" or "3 days or more").

When presented with hypothetical situations representing acute emergency conditions, nearly all household survey respondents stated they would seek care (Table 3). There was variation between situation severity and the timing of care seeking between scenarios.

For households who had experienced at least one emergency condition in the past year, the majority experienced only 1 emergency (Table 4). A little less than half of those with emergencies (47%) experienced a delay in access to care. The majority sought care in a hospital, with most traveling via motorcycle or private care.

There was an association noted between a delay in healthcare access and the insurance of household member 1 (which may be different from the person experiencing the emergency condition) (Table 5). Those with delayed access to healthcare had a higher proportion of individuals who used personal cash and a lower proportion of individuals with national health insurance, with similar proportions of individuals with no or 'other' insurance (p = 0.02). No statistically significant associations were noted based on district, physical accessibility of house, number of household members, or monthly household income.

**Table 3. Emergency situations.**

|  | Scenario 1 (N = 539) | Scenario 2 (N = 539) | Scenario 3 N = 539) | Scenario 4 (N = 539) | Scenario 5 (N = 539) |
|---|---|---|---|---|---|
| **Situation severity** | | | | | |
| Not at all severe | 0 (0.0%) | 2 (0.4%) | 2 (0.4%) | 4 (0.7%) | 3 (0.6%) |
| Possibly severe | 5 (0.9%) | 3 (0.6%) | 0 (0.0%) | 2 (0.4%) | 1 (0.2%) |
| Somewhat severe | 61 (11.3%) | 6 (1.1%) | 7 (1.3%) | 15 (2.8%) | 4 (0.7%) |
| Severe | 156 (28.9%) | 121 (22.4%) | 121 (22.4%) | 202 (37.5%) | 99 (18.4%) |
| Very severe | 317 (58.8%) | 407 (75.5%) | 409 (75.9%) | 316 (58.6%) | 432 (80.1%) |
| **Would you seek care** | | | | | |
| No | 0 (0.0%) | 0 (0.0%) | 0 (0.0%) | 1 (0.2%) | 2 (0.4%) |
| Yes | 539 (100.0%) | 539 (100.0%) | 539 (100.0%) | 538 (99.8%) | 537 (99.6%) |
| **Where would you seek care*** | | | | | |
| Clinic | 5 (0.9%) | 3 (0.6%) | 2 (0.4%) | 2 (0.4%) | 0 (0.0%) |
| Dispensary | 37 (6.9%) | 53 (9.9%) | 38 (7.1%) | 44 (8.2%) | 15 (2.8%) |
| Faith healer | 1 (0.2%) | 1 (0.2%) | 1 (0.2%) | 0 (0.0%) | 0 (0.0%) |
| Health center | 50 (9.3%) | 39 (7.2%) | 55 (10.2%) | 62 (11.5%) | 34 (6.3%) |
| Hospital | 441 (81.8%) | 436 (81.1%) | 442 (82.0%) | 427 (79.4%) | 487 (90.7%) |
| Small shop/pharmacy over-the-counter medications | 5 (1.0%) | 5 (1.0%) | 1 (0.2%) | 3 (0.6%) | 1 (0.2%) |
| Traditional healer | 0 (0.0%) | 1 (0.2%) | 0 (0.0%) | 0 (0.0%) | 0 (0.0%) |
| **How quickly would you seek care*** | | | | | |
| Immediately | 417 (77.4%) | 483 (89.6%) | 498 (92.4%) | 462 (85.9%) | 518 (96.6%) |
| Less than a day | 73 (13.5%) | 47 (8.7%) | 38 (7.1%) | 60 (11.2%) | 16 (3.0%) |
| Between 1–2 days | 39 (7.2%) | 7 (1.3%) | 3 (0.6%) | 16 (3.0%) | 2 (0.4%) |
| 3 or more days | 10 (1.9%) | 2 (0.4%) | 0 (0.0%) | 0 (0.0%) | 0 (0.0%) |

*Missing values: Where would you seek care (scenario 2=1; scenario 4=1; scenario 5=2); How quickly would you seek care (scenario 4=1; scenario 5=3).

**Table 4. Access to treatment of emergency conditions by limited/delayed emergency care.**

| | No delayed access (N = 40) | Delayed access (N = 36) | Total (N = 76) |
|---|---|---|---|
| **Any household members with an emergency in the last year** | 40 (100.0%) | 36 (100.0%) | 76 (100.0%) |
| **How many emergencies** | | | |
| 1 | 36 (90.0%) | 29 (80.6%) | 65 (85.5%) |
| 2 | 3 (7.5%) | 6 (16.7%) | 9 (11.8%) |
| 3 | 1 (2.5%) | 1 (2.8%) | 2 (2.6%) |
| **Did any of these people die** | 3 (7.5%) | 1 (2.8%) | 4 (5.3%) |
| **Which household member had an emergency**[*] | | | |
| Total emergencies | 45 | 44 | 89 |
| 1st oldest household member (#1) | 18 (40.0%) | 18 (40.9%) | 36 (40.4%) |
| 2nd oldest household member (#2) | 14 (31.1%) | 13 (29.5%) | 27 (30.3%) |
| 3rd oldest household member (#3) | 1 (2.2%) | 6 (13.6%) | 7 (7.9%) |
| 4th oldest household member (#4) | 4 (8.9%) | 1 (2.3%) | 5 (5.6%) |
| 5th oldest household member (#5) | 2 (4.4%) | 2 (4.5%) | 4 (4.5%) |
| 6th oldest household member (#6) | 1 (2.2%) | 0 (0.0%) | 1 (1.1%) |
| 7th oldest household member (#7) | 2 (4.4%) | 0 (0.0%) | 2 (2.2%) |
| 8th oldest household member (#8) | 1 (2.2%) | 0 (0.0%) | 1 (1.1%) |
| Other household member not listed | 2 (4.4%) | 4 (9.1%) | 6 (6.7%) |
| **What type of emergency**[*] | | | |
| Car or motorcycle | 5 (11.1%) | 3 (6.8%) | 8 (9.0%) |
| Chest pain | 3 (6.7%) | 4 (9.1%) | 7 (7.9%) |
| Confusion | 4 (8.9%) | 1 (2.3%) | 5 (5.6%) |
| Difficulty breathing | 9 (20.0%) | 6 (13.6%) | 15 (16.9%) |
| Fall | 10 (22.2%) | 11 (25.0%) | 21 (23.6%) |
| Fever | 3 (6.7%) | 3 (6.8%) | 6 (6.7%) |
| Nausea, vomiting, diarrhea | 0 (0.0%) | 1 (2.3%) | 1 (1.1%) |
| Trouble speaking | 0 (0.0%) | 1 (2.3%) | 1 (1.1%) |
| Penetrating trauma | 1 (2.2%) | 1 (2.3%) | 2 (2.2%) |
| Severe pain | 2 (4.4%) | 8 (18.2%) | 10 (11.2%) |
| Snake/animal/insect bite | 0 (0.0%) | 3 (6.8%) | 3 (3.4%) |
| Other | 8 (17.8%) | 2 (4.5%) | 10 (11.2%) |
| **Did you/they seek care**[*] | 41 (91.1%) | 43 (97.7%) | 84 (94.4%) |
| **Location of care**[*] | | | |
| Dispensary | 5 (12.2%) | 11 (25.6%) | 16 (19.0%) |
| Faith healer | 2 (4.9%) | 0 (0.0%) | 2 (2.4%) |
| Health center | 7 (17.1%) | 4 (9.3%) | 11 (13.1%) |
| Hospital | 24 (58.5%) | 26 (60.5%) | 50 (59.5%) |
| Pharmacy/other | 3 (7.3%) | 2 (4.7%) | 5 (6.0%) |
| **Reason for choice of location**[*] | | | |
| Available specialty | 3 (7.3%) | 3 (7.0%) | 6 (7.1%) |
| Care of specific doctor | 0 (0.0%) | 1 (2.3%) | 1 (1.2%) |
| Closest facility | 25 (61.0%) | 22 (51.2%) | 47 (56.0%) |
| Cost | 2 (4.9%) | 3 (7.0%) | 5 (6.0%) |
| Covered by insurance | 1 (2.4%) | 1 (2.3%) | 2 (2.4%) |
| Familiarity | 2 (4.9%) | 2 (4.7%) | 4 (4.8%) |
| Quality of care | 8 (19.5%) | 11 (25.6%) | 19 (22.6%) |

*(Continued)*

**Table 4.** (Continued)

| | No delayed access (N = 40) | Delayed access (N = 36) | Total (N = 76) |
|---|---|---|---|
| **How long did it take to get 1st health care (hours)*** | | | |
| Mean (SD) | 2.0 (4.1) | 2.5 (5.3) | 2.3 (4.7) |
| **How did you/they get there*** | | | |
| Bajaji | 5 (12.2%) | 5 (11.6%) | 10 (11.9%) |
| Car private | 13 (31.7%) | 15 (34.9%) | 28 (33.3%) |
| Car taxi | 2 (4.9%) | 0 (0.0%) | 2 (2.4%) |
| Dala dala | 3 (7.3%) | 7 (16.3%) | 10 (11.9%) |
| Motorcycle | 12 (29.3%) | 14 (32.6%) | 26 (31.0%) |
| Unknown/can't remember | 0 (0.0%) | 1 (2.3%) | 1 (1.2%) |
| Walking | 6 (14.6%) | 1 (2.3%) | 7 (8.3%) |

*Statistics based off the total number of emergencies, with some households having multiple members with an emergency.

**Table 5.** Delayed access to healthcare.

| | No delayed access (N = 40) | Delayed access (N = 36) | Total (N = 76) | p-value |
|---|---|---|---|---|
| **District of residence** | | | | 0.13 |
| Hai | 15 (37.5%) | 14 (38.9%) | 29 (38.2%) | |
| Moshi municipal | 13 (32.5%) | 5 (13.9%) | 18 (23.7%) | |
| Moshi district | 12 (30.0%) | 17 (47.2%) | 29 (38.2%) | |
| **Difficult accessibility of house*** | | | | 0.52 |
| No | 7 (17.5%) | 4 (11.1%) | 11 (14.5%) | |
| Yes | 33 (82.5%) | 32 (88.9%) | 65 (85.5%) | |
| **Insurance status household member 1** | | | | 0.022 |
| None | 4 (10.0%) | 4 (11.1%) | 8 (10.5%) | |
| Cash personal | 17 (42.5%) | 26 (72.2%) | 43 (56.6%) | |
| National Health Insurance | 16 (40.0%) | 6 (16.7%) | 22 (28.9%) | |
| Other | 3 (7.5%) | 0 (0.0%) | 3 (3.9%) | |
| **Number of household members** | | | | 0.90 |
| Median (Q1, Q3) | 4.5 (3.0, 6.0) | 5.0 (3.5, 6.0) | 5.0 (3.0, 6.0) | |
| **Monthly household income** | | | | 0.63 |
| Median (IQR) | 225000 (150000, 325000) | 200000 (100000, 300000) | 200000 (100000, 310000) | |

*Difficult accessibility of house was defined as a household on a non-paved road or a household that could not be reached by a four-wheel vehicle.

Statistically significant associations with inappropriate emergency response(s) were noted for several variables (Table 6). A higher proportion of those with inappropriate response(s) lived in the Hai district (36.0% vs 19.7%), while a lower proportion lived in the Moshi Urban district (14.2% vs 35.7%), with similar proportions living in Moshi Rural when compared to those with appropriate responses to all questions (p < 0.001). A higher proportion of those with an inappropriate response(s) was uninsured (11.1% vs 4.1%) and a lower proportion had national health insurance (20.0% vs 28.0%), with similar proportions using personal cash or 'other' insurance when compared to those with appropriate responses to all questions (p = 0.005). Those with inappropriate response(s) had a lower monthly household income (median 140,000 vs 200,000 Tanzanian shillings) when compared to those with appropriate responses to all questions (p < 0.001).

**Table 6. Inappropriate emergency response.**

| | No (N = 314) | Yes (N = 225) | Total (N = 539) | p-value |
|---|---|---|---|---|
| **Any household members with an emergency in the last year** | | | | 0.053 |
| No | 262 (83.4%) | 201 (89.3%) | 463 (85.9%) | |
| Yes | 52 (16.6%) | 24 (10.7%) | 76 (14.1%) | |
| **District of residence** | | | | <0.001 |
| Hai | 62 (19.7%) | 81 (36.0%) | 143 (26.5%) | |
| Moshi Municipal | 112 (35.7%) | 32 (14.2%) | 144 (26.7%) | |
| Moshi District | 140 (44.6%) | 112 (49.8%) | 252 (46.8%) | |
| **Insurance status household member 1** | | | | 0.005 |
| None | 13 (4.1%) | 25 (11.1%) | 38 (7.1%) | |
| Cash personal | 207 (65.9%) | 152 (67.6%) | 359 (66.6%) | |
| National Health Insurance | 88 (28.0%) | 45 (20.0%) | 133 (24.7%) | |
| Other | 6 (1.9%) | 3 (1.3%) | 9 (1.7%) | |
| **Number of household members** | | | | 0.29 |
| Median (Q1, Q3) | 4 (3, 6) | 4 (3, 5) | 4 (3, 6) | |
| **Monthly household income** | | | | <0.001 |
| Median (Q1, Q3) | 200000 (100000, 320000) | 140000 (80000, 250000) | 150000 (90000, 300000) | |

While not statistically significant at the 0.05 level of significance, those with inappropriate medical responses had a lower proportion of household emergencies in the previous year (10.7% vs 16.6%) when compared to those with appropriate responses to all questions (p = 0.053). No association was noted between the number of family members and inappropriate response(s).

## Discussion

The findings from this community-based survey done in Northern Tanzania provide critical insight into the emergency care system and potential areas for improvement. Our findings suggest that there are significant challenges in accessing the emergency care system in Northern Tanzania. Our findings highlight the need to improve the emergency care system by expanding coverage of health care insurance for all to reduce the burden of cost, increasing communities' education and awareness of symptoms for common emergencies, and developing of emergency medical services (EMS) to improve pre-hospital care.

In LMICs, health financing is a significant barrier to accessing care. In this study, the lack of health insurance or the need to pay in cash was associated with delayed health care access. Those without insurance were also more likely to have an inappropriate response to the theoretical emergency scenarios, suggesting that either there was a lack of understanding of what constitutes an emergency condition, or that the potential financial cost would impact any decisions to seek emergency care in the future. This is reflected in another study which assessed financial toxicity due to out-of-pocket costs in patients who presented to the emergency department at Kilimanjaro Christian Medical Center [17]. In high-income countries, the issue is often related to the high cost of healthcare and lack of financial protection, whereas in low-income countries, including countries in sub-Saharan Africa, the lack of comprehensive insurance coverage and limited financial resources and savings exacerbate the problem [18]. These findings highlight the importance of health financing strategies in strengthening the emergency care system by reducing the financial burden due to out-of-pocket expenditure [19,20]. Potential strategies could focus increasing National Health Insurance Fund coverage

to households with lower incomes or by further development of low-cost or free prehospital transport for patients experiences emergency medical conditions.

While there are similarities in the financial challenges faced by individuals without insurance or those paying cash in both high-income and low-income countries, it is important to note that the underlying factors and implications can differ significantly. Efforts to address these challenges should focus on improving access to affordable healthcare services, expanding insurance coverage, and implementing policies that alleviate the financial burden on individuals seeking care. Such measures can help reduce delays in seeking care and promote better health outcomes for individuals in low-income countries [21].

Insufficient knowledge of emergency symptoms can pose significant barriers to seeking timely medical care. A lack of awareness may impede individuals from recognizing the severity of symptoms leading to delayed decision-making in seeking care for emergency conditions. This delay to seek care can be particularly critical in time-sensitive conditions where prompt intervention is crucial for better outcomes. A recent study in Singapore by Quah et al. on heart attack symptoms and chest pain found that a more educated population with higher socioeconomic status had more knowledge of emergency signs and symptoms compared to those with low education, those residing in rural areas, and those with low income [22,23]. These findings are similar to those in other high-income countries [24,25]. A recent study of emergency department patients in rural Uganda also noted potential cultural factors may also contribute to delays to seek care [26]. A study conducted in Northern Tanzania on barriers to acute coronary syndrome services participants also highlighted education as an issue where some of the patients would link their symptoms to superstitious beliefs instead of seeking medical care [27].

In this study, we observe, when asked about hypothetical emergencies, respondents who had an inappropriate response to the five emergency scenarios were more likely to be uninsured, had a lower income, and resided in rural areas. These socioeconomic factors often can exacerbate knowledge gaps and compound existing challenges in health care efficiently [28–30]. Leveraging existing local community health organizations in South Africa and Zambia has proven to be a successful approach in disseminating critical education during the COVID-19 pandemic [31]. A similar approach to educate the community regarding emergency conditions using trusted resources embedded in the community may prove to be effective.

Distance from health facilities as a population-related factor also affects the utilization of emergency care. This is a significant problem in Africa, where nearly 30% of the population lives greater than 2 hours travel time from the nearest healthcare facility [32]. We found that participants residing in rural districts had higher percentages of delays compared to their urban counterparts, although these findings were not statistically significant. When assessing understanding through hypothetical scenarios, we found that the majority of participants inhabiting areas other than Moshi Urban where emergency care services are available had inappropriate responses compared to Moshi urban counterparts. Other studies have found that distance may affect healthcare utilization through the time that transportation takes as well as factors such as availability of means of transport, road access, and cost of transportation [33–35]. A recent study in Burkina Faso assessing child mortality found that increased distance from healthcare was associated with decreased care seeking [36]. Two recent studies using the three-delays model to assess factors contributing to delays in care in the Kilimanjaro region also highlighted distance as a factor that affects both healthcare-seeking and emergency condition recognition [9,37]. This demonstrates a further need for scaling up emergency care services to primary health care facilities, capacity building, and infrastructure development. Additionally, there is a need for the development of prehospital care, an important component of emergency care services, through either structured ambulance

services or utilizing existing community-based transportation, such as may have been developed for obstetric emergencies [38]. A similar approach has proven to be effective in Rwanda through strengthening the capacity of emergency care services which showed improved overall outcomes [39,40].

Another approach is to improve the capacity of local health centers to manage emergency conditions, therefore increasing the number of access points for individuals seeking emergency care. The Tanzanian government recently scaled up Emergency Care Services through constructing, retooling, and revamping emergency departments from primary to tertiary healthcare facilities throughout Tanzania. This was done through the COVID-19 Response and Recovery Plan where 80 Emergency Medicine Departments (EMDs) were constructed in primary health care facilities, and 35 in secondary health care facilities (regional hospitals). Additionally, 22 EMDs were renovated in regional hospitals and 4 EMDs were retooled in zonal tertiary hospitals [41]. This represents good progress to closing the infrastructural gap. However, patient-centered barriers remain, such as the need for human resources as well as a lack of community education on emergency care services. There is a need for multi-sectoral involvement including integration of emergency condition education in school curriculums and ensuring adequate road access.

## Limitations

This study had several limitations. First, due to inaccurate census data, we had to change our sampling strategy early in data collection. This change in sampling strategy may have resulted in a selection bias given that 30 villages used the initial strategy. Given the long distances that research assistants were required to walk in order to find a household with the initial strategy, this may have resulted in a non-randomized sample thus impacting the overall representation of the sample. Secondly, we limited survey questions to the head of household, who then responded for all of the other individuals within their household. This may have resulted in some recall bias leading to inaccurate or missing information as the survey participant was not necessarily the individual who experienced the emergency. Next, when asking survey respondents for information regarding emergency conditions over the last year, we did not define what this entailed, thus leaving it open to the respondent's interpretation. Participants may have had a different interpretation of what constitutes an emergency and language barrier, especially for those non-primary Kiswahili speakers. However, only patients who were able to speak Kiswahili fluently were included in the study. Finally, paper forms were used in settings when connectivity was limited, these data were then transferred to RedCap once connectivity was restored. This process of data transfer may have resulted in data misclassification. To minimize this risk, data were carefully reviewed by members of the study team and weekly quality meetings were conducted to ensure accurate data collection.

## Conclusion

Delayed access to care for emergency conditions in Northern Tanzania is impacted by gaps in understanding emergency care, distance to the healthcare facilities, cost, and lack of insurance. Efforts to address the challenges identified in this study should focus on improving access, including a comprehensive public health education campaign to improve health literacy. Basic emergency care education should be incorporated in school curriculums, social media and mass media health promotion programs. Additionally, there should be a focus on increasing affordable healthcare services by expanding insurance coverage, development of emergency medical services and implementing policies that alleviate the financial burden on individuals seeking care. Such measures can help reduce delays in seeking care and promote better health

outcomes. In addition, community education and engagement in recognition of emergency symptoms may further contribute to the reduction of delays in seeking care as increased awareness empowers individuals to act promptly and timely hence preventing complications and fostering overall well-being.

## Supporting information

**S1 File. Community needs assessment survey.**
(DOCX)

**S1 Checklist. Inclusivity in Global Health Research questionnaire.**
(DOCX)

## Acknowledgments

We would like to acknowledge our Moshi-based study team (Stephen Sikumbili, Yvonne Sawe, Lazaro Arangare, Flora Nkya) and the KCRI/Duke Collaboration.

## Author contributions

**Conceptualization:** Francis Sakita, Joao Ricardo Nickenig Vissoci, Blandina T. Mmbaga, Catherine A. Staton, Anjni P. Joiner.

**Formal analysis:** Thiago Rocha Hernandes, Joao Ricardo Nickenig Vissoci, Alexander Gordee, Maragatha Kuchibhatla, Linda Minja.

**Funding acquisition:** Anjni P. Joiner.

**Investigation:** Anjni P. Joiner.

**Methodology:** Thiago Rocha Hernandes, Joao Ricardo Nickenig Vissoci, Linda Minja, Catherine A. Staton, Anjni P. Joiner.

**Project administration:** Francis Sakita.

**Supervision:** Francis Sakita, Maragatha Kuchibhatla, Blandina T. Mmbaga, Catherine A. Staton, Anjni P. Joiner.

**Writing – original draft:** Frida Shayo, Gregory Goodluck Zaccheus, Michael Kiremeji, Elizabeth M. Keating, Anjni P. Joiner.

**Writing – review & editing:** Frida Shayo, Gregory Goodluck Zaccheus, Francis Sakita, Joao Ricardo Nickenig Vissoci, Alexander Gordee, Maragatha Kuchibhatla, Michael Kiremeji, Linda Minja, Blandina T. Mmbaga, Catherine A. Staton, Elizabeth M. Keating, Anjni P. Joiner.

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
