## [Decision Letter · Decision Letter 0]

27 Aug 2024

PGPH-D-24-01792

Emergency care knowledge, utilization, and barriers in Northern Tanzania: A community-based survey

Dear Dr. Joiner,

Thank you for submitting your manuscript to PLOS Global Public Health. After careful consideration, we feel that it has merit but does not fully meet PLOS Global Public Health’s publication criteria as it currently stands. Therefore, we invite you to submit a revised version of the manuscript that addresses the points raised during the review process.

The reviewers have provided valuable comments especially on the methods and discussion section which would help to improve the quality of your submission. Hence, I invite you to respond to these reviewer comments. 

We look forward to receiving your revised manuscript.

Kind regards,

Mathew Sunil George

Academic Editor

Journal Requirements:

Additional Editor Comments (if provided):

Reviewers' comments:

Reviewer's Responses to Questions

**Comments to the Author**

1. Does this manuscript meet PLOS Global Public Health’s publication criteria ? Is the manuscript technically sound, and do the data support the conclusions? The manuscript must describe methodologically and ethically rigorous research with conclusions that are appropriately drawn based on the data presented.

Reviewer #1: Yes

Reviewer #2: Yes

Reviewer #3: Yes

2. Has the statistical analysis been performed appropriately and rigorously?

Reviewer #1: Yes

Reviewer #2: Yes

Reviewer #3: Yes

3. Have the authors made all data underlying the findings in their manuscript fully available (please refer to the Data Availability Statement at the start of the manuscript PDF file)?

Reviewer #1: No

Reviewer #2: No

Reviewer #3: Yes

4. Is the manuscript presented in an intelligible fashion and written in standard English?

Reviewer #1: Yes

Reviewer #2: Yes

Reviewer #3: Yes

5. Review Comments to the Author

Reviewer #1: Thank you for allowing me to review this abstract. Overall, this is a very important household survey assessing health literacy and access to emergency care.

Abstract: background or results, consider adding additional descriptive data about the community that was interviewed as well as the health system eg income brackets, rural vs urban, number of facilities and/or distance to facility

Methods:

line 100 “The three districts share a total of 13 hospitals” Please provide more details about the types and distribution of facilities, is there any data on proximity to facilities for the community. What types of services are available at each level/type of facility? Many readers will not understand tiered systems and these can vary significantly by region in terms of resources.

Line 125: how were households randomly sampled?

The headers in methods are a bit confusing, sample size and sampling could be split into sample size and then a new section combining the sampling techniques, labeled as such (rather than study protocols) eg Selection of households

Line 171: Why snakebites, I would assume the survey was multi-purpose, briefly explain this

Line 175: “traumatic situations” potentially unclear as trauma can be injury, emotional etc, suggestion for “injuries”

Lines 180-182: Where respondents choosing from a list of options? Or were free text options available

Results Table 2: What is the rationale behind reporting tribe and religion, if this does nto add to the discussion or explanation, would consider deleting

Number of household members consider presenting either median (IQR) or mean Q1, Q3, you do not need 4 lines representing the same data, remove missing, report when significant

Monthly household income, same comment as household members, report one value and associated measure of variance

Residence, only need yes or no line, not both

Table 3: missing data can be supplied as text or footnotes to table, row is very distracting in the table

Line 243: How is delayed access defined?

Table 4: overall this table is too big, consider adding header rows or dividing into multiple tables

consider adding a row above which household member had an emergency with N that can then carry though for the all the following sections

“how long did it take…” present one measure and it’s variance in same row

line 251: confusing sentence, “individuals with cash personal”

Table 5:

Difficult accessibility of house-how is this defined?

Why is number of household members included here, one line with measure and variance

Income, suggest one measure with variance in one row, add row with USD conversions for global audience

Table 6: same comments as previously listed, decrease number of rows with variables containing mean, median, Q1, Q3 and range

Line 268-271, confusing sentence, please clarify

Discussion: lines 304-321: the points made are repetitive in these 2 paragraphs. what do the authors recommend in response to this association, is there evidence for successful interventions that improve health literacy

Lines 335-339 terms emergency care services and prehospital care are used interchangeably, they are not the same. Why is an ambulance the only potential solution? There have been successful community transport initiatives, mostly in the OB literature

The final discussion points on expanding emergency care at facilities and expanding prehospital care are intertwined and may be confusing to non-EM providers. Both are important but this is not entirely clear as written

Limitations

Line 365: this is the first time that potential language barriers are mentioned, what actions were taken to overcome this, how is the language barrier linked to interpretation of an emergency?

Reviewer #2: The study addresses a critical need for improving emergency systems in low-resource settings and makes a substantial contribution to public health literature in this area.

However, there are areas that require clarification and further elaboration, particularly in the methods, results, and discussion sections. The manuscript would benefit from clearer explanations of the sampling strategy, more detailed context regarding income levels and their impact on emergency care, and expanded discussion on policy recommendations for improving emergency care systems in similar settings. Please find attached comments.

Reviewer #3: The authors adequately explain the importance of the article within its field.

The methodology is sound, even considering the multiple sampling strategies due to logistics.

Considerations:

Lines 358 to 360: The survey was conducted with the head of household. Culturally, in Northern Tanzania, is the head of household the medical decision maker? For example, in Togo, head of household is often a male but decisions regarding medical need for any family members is managed by the household manager, usually the mother or another older woman. Could a mismatch be a limitation in this study? (Lines 154-155: In methods, informed consent was from both HoH and health care decision maker, but it is not clear if the health care decisionmaker performed the interview)

Line 359: Responded “to”? Or responded “for”?

Line 370: The actual conclusion of the study is not listed in the conclusion section of the manuscript (though it can be seen in the abstract). The next steps listed are appropriate and highly relevant.

Lines 364 to 365: Are there other studies (likely outside the direct emergency medicine sector) in Tanzania or surrounding areas, that have investigated the understanding of emergencies? The language or culture of an emergency, perhaps anthropologically? Though this could be limited, such information could add context and depth for this and future research.

References - 60% of the supporting articles were published in the past 7 years with many studies from geographically proximal areas which improves relevance.

6. PLOS authors have the option to publish the peer review history of their article (what does this mean? ). If published, this will include your full peer review and any attached files.

**Do you want your identity to be public for this peer review?** For information about this choice, including consent withdrawal, please see our Privacy Policy .

Reviewer #1: No

Reviewer #2: **Yes: ** Khushbu Balsara

Reviewer #3: No

---

## [Decision Letter · Decision Letter 1]

21 Nov 2024

Emergency care knowledge, utilization, and barriers in Northern Tanzania: A community-based survey

PGPH-D-24-01792R1

Dear Dr Joiner,

We are pleased to inform you that your manuscript 'Emergency care knowledge, utilization, and barriers in Northern Tanzania: A community-based survey' has been provisionally accepted for publication in PLOS Global Public Health.

Best regards,

Mathew Sunil George

Academic Editor

Reviewer Comments (if any, and for reference):

Reviewer's Responses to Questions

**Comments to the Author**

1. If the authors have adequately addressed your comments raised in a previous round of review and you feel that this manuscript is now acceptable for publication, you may indicate that here to bypass the “Comments to the Author” section, enter your conflict of interest statement in the “Confidential to Editor” section, and submit your "Accept" recommendation.

Reviewer #1: (No Response)

Reviewer #2: All comments have been addressed

2. Does this manuscript meet PLOS Global Public Health’s publication criteria ? Is the manuscript technically sound, and do the data support the conclusions? The manuscript must describe methodologically and ethically rigorous research with conclusions that are appropriately drawn based on the data presented.

Reviewer #1: Yes

Reviewer #2: Yes

3. Has the statistical analysis been performed appropriately and rigorously?

Reviewer #1: Yes

Reviewer #2: Yes

4. Have the authors made all data underlying the findings in their manuscript fully available (please refer to the Data Availability Statement at the start of the manuscript PDF file)?

Reviewer #1: Yes

Reviewer #2: Yes

5. Is the manuscript presented in an intelligible fashion and written in standard English?

Reviewer #1: Yes

Reviewer #2: Yes

6. Review Comments to the Author

Reviewer #1: Authors have made adequate changes.

In Table 4, removing the yes/no lines from some of the questions, has made it unclear if the question was answered, for example, "Did any one die" needs a yes/no label, this occurs for several rows in table 4.

Reviewer #2: The authors have addressed previous comments and questions.

7. PLOS authors have the option to publish the peer review history of their article (what does this mean? ). If published, this will include your full peer review and any attached files.

**Do you want your identity to be public for this peer review?** For information about this choice, including consent withdrawal, please see our Privacy Policy .

Reviewer #1: No

Reviewer #2: No
